# Webscale-RL: Automated Data Pipeline for Scaling RL Data to Pretraining Levels

**Zhepeng Cen**[1,2], **Haolin Chen**[1], **Shiyu Wang**[1], **Zuxin Liu**[1], **Zhiwei Liu**[1],
**Ding Zhao**[2], **Caiming Xiong**[1], **Huan Wang**[1], **Weiran Yao**[1]

[1]Salesforce AI Research, [2]Carnegie Mellon University

## Abstract

Large Language Models (LLMs) have achieved remarkable success through imitation learning on vast text corpora, but this paradigm creates a training-generation gap and limits robust reasoning. Reinforcement learning (RL) offers a more data-efficient solution capable of bridging this gap, yet its application has been constrained by a critical data bottleneck: existing RL datasets are orders of magnitude smaller and less diverse than web-scale pre-training corpora. To address this, we introduce the **Webscale-RL pipeline**, a scalable data engine that systematically converts large-scale pre-training documents into millions of diverse, verifiable question-answer pairs for RL. Using this pipeline, we construct the **Webscale-RL dataset**, containing 1.2 million examples across more than 9 domains. Our experiments show that the model trained on this dataset significantly outperforms continual pretraining and strong data refinement baselines across a suite of benchmarks. Notably, RL training with our dataset proves substantially more efficient, achieving the performance of continual pre-training with up to $100\times$ fewer tokens. Our work presents a viable path toward scaling RL to pretraining levels, enabling more capable and efficient language models.

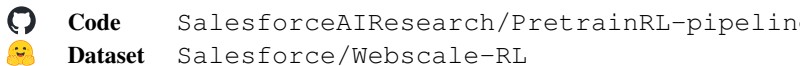

|  | Code | SalesforceAIResearch/PretrainRL-pipeline |
|---|---|---|
|  | Dataset | Salesforce/Webscale-RL |

## 1 Introduction

Large Language Models (LLMs) have achieved remarkable success, primarily through learning on vast text corpora. However, this predominant paradigm, which includes pretraining with next-token prediction and supervised fine-tuning (SFT), is fundamentally in the form of imitation learning. By training models to mimic static offline datasets, imitation learning creates a "teacher-forcing" dependency that makes models vulnerable to distribution shifts (Ross et al., 2011; Levine et al., 2020; Foster et al., 2024) and leads to a significant gap between training and generation dynamics (Chen et al., 2024c; Bachmann & Nagarajan, 2024; Cen et al., 2024). Consequently, models trained in this way struggle with distribution shift and lack the robust reasoning abilities required for complex problem solving.

Reinforcement learning (RL) offers a powerful alternative to overcome these challenges (Shao et al., 2024; DeepSeek-AI et al., 2025). By learning from online reward feedback on its own generations, an RL-trained model can explore a wider solution space and is not confined to a static dataset, bridging the training-inference gap. This online learning process makes RL a significantly more data-efficient training paradigm. As our empirical results demonstrate, RL can achieve performance gains comparable to continual pretraining with up to two orders of magnitude fewer tokens, providing a compelling motivation for scaling RL to unlock new levels of model capability and efficiency.

Despite the clear advantages of RL, its adoption at scale is severely hampered by a critical data bottleneck and most existing practice in RL training is mainly limited to reasoning tasks such as math and coding in the post-training stage. Pretraining corpora are measured in trillions of tokens, whereas existing RL datasets are orders of magnitude smaller (e.g., <10B tokens for RL vs. >1T tokens for pretraining) and lack the diversity of web-scale data. This scarcity is driven by the high

Figure 1: The scaling on LLM RL is fundamentally bottlenecked by the scarcity of high-quality RL data. While pretraining leverages >1T diverse web tokens, RL datasets remain limited to <10B tokens with limited diversity. We propose `Webscale-RL` data pipeline to fundamentally improve the scalability of RL data: we convert the pretraining corpora to verifiable query and ground-truth answer pairs, scaling RL data to pretraining levels while preserving the diversity. The experiments show that RL with `Webscale-RL` data is significantly more effective and efficient than continual pretraining and data refinement baselines.

cost of generating high-quality, verifiable question-answering (QA) pairs, which are essential for effective RL-based reasoning tasks. This immense disparity in data scale and diversity prevents RL from realizing its full potential to enhance the general reasoning capabilities of LLMs.

To address these limitations, we introduce **Webscale-RL**, a scalable data pipeline that systematically converts large-scale pretraining corpora into massive, diverse, and verifiable RL-ready datasets. Our pipeline is designed to bridge the data gap between pretraining and reinforcement learning, unlocking the potential to train LLMs with RL at a scale previously unattainable while preserving the vast diversity of the original pretraining data.

Our main contributions are threefold:

- We propose **Webscale-RL pipeline**, an automated and scalable data engine that converts web-scale pretraining documents into verifiable question-answer pairs for RL. The pipeline incorporates stages for data filtering, domain and persona-driven generation, and quality verification to ensure high-quality output.
- We construct **Webscale-RL dataset**, a large-scale and diverse RL dataset containing 1.2 million verifiable QA pairs spanning over nine domains. Our analysis shows it is significantly more diverse than existing large-scale RL datasets.
- We provide empirical evidence demonstrating the effectiveness of our approach. The model trained with RL on the `Webscale-RL` dataset significantly outperforms continual pretraining on the source data and strong data refinement baselines across a wide range of benchmarks. Furthermore, our results show that RL with our dataset is substantially more data-efficient, achieving comparable performance to continual pretraining with $100\times$ fewer tokens.

Our work demonstrates that by converting massive pretraining corpora into a format suitable for RL, we can unlock significant performance and efficiency gains. This provides a path toward scaling reinforcement learning to match the scale of pretraining, leading to a new generation of more capable and robust language models.

## 2 RELATED WORKS

**Training Data Development and Synthesis**. The development of LLMs hinges on the availability of vast, high-quality training datasets. However, curating such corpora, especially labeled and across diverse domains, is often prohibitively expensive and time-consuming. This challenge has spurred significant research into efficient synthetic data generation pipelines (Chen et al., 2024a; Ma et al., 2025; Fan et al., 2025). Current pre-training corpora are typically compiled from a variety of public sources, such as Wikipedia (Foundation), large-scale web crawls (Computer, 2023; Paster et al., 2023; Penedo et al., 2024a; Li et al., 2024; Raffel et al., 2019), and code repositories (Lozhkov et al., 2024). This is often supplemented with content from digitized books (Stroube, 2003) and data

generated by other LLMs (Ben Allal et al., 2024; Huang et al., 2024). To endow LLMs with comprehensive knowledge and robust downstream capabilities, these corpora are constructed on a massive scale, often containing tens of trillions of tokens, as exemplified by the 30-trillion-token RedPajama dataset (Computer, 2023) and the 67.5-trillion-token Stack-v2 (Lozhkov et al., 2024). More recently, the success of models like DeepSeek-R1-Zero (DeepSeek-AI et al., 2025) and DeepSeek-R1-Zero (DeepSeek-AI et al., 2025) and Grok-4 (xAI, 2025), which integrate reinforcement learning (RL) at the pre-training stage, is built on and is built on and has intensified the demand for similarly large and high-quality synthetic synthetic RL datasets. There have been multiple lines of work in large-scale data synthesis for LLM training: DeepScaler (Luo et al., 2025) curated a small (40K) RL dataset for mathematical reasoning; OpenR1-Math (Hugging Face, 2025) further scales up the mathematical RL dataset for both SFT and RL via distillation, resulting in a 220K dataset; WebInstruct (Yue et al., 2024), OpenThoughts (Guha et al., 2025) and NatureReasoning (Yuan et al., 2025) expand the distillation path to multiple domains and synthesize over 1M data using teacher models for SFT, respectively; Nemotron (Bercovich et al., 2025) extends the dataset size to 3.9M for both SFT and RL.

**Reinforcement Learning in LLMs**. Recent advancements in large-scale RL have significantly enhanced the capabilities of LLMs, as demonstrated by models such as OpenAI's o-series (OpenAI, 2024a; Jaech et al., 2024; OpenAI, 2024b) and DeepSeek-V3/R1 (DeepSeek-AI et al., 2024; 2025). Besides these models, many other works show that LLMs trained to reason with Chain-of-Thought (CoT) prompting have shown substantial performance gains in diverse areas, including mathematical and scientific reasoning (Xie et al., 2025; Cen et al., 2025; Shao et al., 2024; Luong et al., 2024; Chen et al., 2024b; Cui et al., 2025), code generation (Le et al., 2022; Wei et al., 2025), and tool use (Zhang et al., 2025a; Qian et al., 2025). The optimization of the RL objective in these models is primarily driven by foundational algorithms like Proximal Policy Optimization (PPO) (Schulman et al., 2017) and its variant, Group Relative Policy Optimization (GRPO) (Shao et al., 2024). Rooted from post-training RL, many works further extend RL to a significantly larger scale (Liu et al., 2025c;b; xAI, 2025) or an earlier stage like pre-training (Zelikman et al., 2024; Dong et al., 2025; Li et al., 2025), indicating the effectiveness of prolonged, large-scale RL training.

## 3 METHODOLOGY

In this section, we first provide a brief comparison of the pretraining and RL training paradigms and then present our `Webscale-RL` data pipeline that systematically converts large-scale pretraining data into RL data while preserving the scale and diversity of web data.

### 3.1 PRELIMINARIES

**Pretraining**. In the pretraining stage, a large-scale corpus $\mathcal{D}_{\text{pretraining}}$ (usually $>$1T tokens) is constructed by filtering and deduplicating publicly available web data sources (Penedo et al., 2024b; Weber et al., 2024; Li et al., 2024). Given this static dataset, the LLM is trained in a teacher-forcing manner to imitate the next-token distribution of the data by minimizing the negative log-likelihood:

$$\min_{\theta} -\mathbb{E}_{\mathbf{x} \sim \mathcal{D}_{\text{pretraining}}} \left[ \sum_{t=1}^{T} \log P_{\theta}(x_t \mid \mathbf{x}_{(<t)}) \right], \tag{1}$$

where $\mathbf{x} = [x_1, \ldots, x_T]$ is a token sequence sampled from the pretraining dataset $\mathcal{D}_{\text{pretraining}}$. This imitation-based objective enforces the model to learn the given pattern from the demonstration data but does not expose the model to the distribution induced by its own generations, suffering from the distribution shift issue (Bachmann & Nagarajan, 2024; Ross et al., 2011) and leading to a training-inference gap (Bengio et al., 2015; Levine et al., 2020).

**Reinforcement Learning (RL)**. RL instead optimizes the model as a policy that *generates answers online* and maximizes expected reward on a query $\mathbf{q}$:

$$\max_{\theta} \mathbb{E}_{\mathbf{q} \sim Q, \mathbf{a} \sim P_{\theta}(\cdot|\mathbf{q})} \left[ R(\mathbf{q}, \mathbf{a}) \right], \tag{2}$$

where $Q$ is the query set and $R$ is a task-specific reward function. The online generation and feedback loop enable the model to narrow the training-inference gap. In our setup, we adopt a binary reward that returns 1 only when the model's final answer matches the ground-truth answer and 0 otherwise. Consequently, each RL training instance is a verifiable question-answer pair.

## 3.2 WEBSCALE-RL DATA PIPELINE

While RL has shown promise in enhancing LLM capabilities (Jaech et al., 2024; DeepSeek-AI et al., 2025), its effectiveness is constrained by the limited scale and diversity of existing RL datasets. Therefore, the RL training is typically conducted on a much smaller scale on limited domains in the post-training stage. This discrepancy arises from the high costs associated with human annotation and the challenges in generating verifiable QA pairs at large scale. Furthermore, most existing RL datasets focus on specific tasks or domains and thus lack the breadth of topics and styles found in web-scale corpora, limiting their generalization to diverse real-world scenarios. To address the limited volume and diversity of existing RL datasets, we propose a `Webscale-RL` data pipeline that converts pretraining documents into RL data at scale while preserving the diversity of web data.

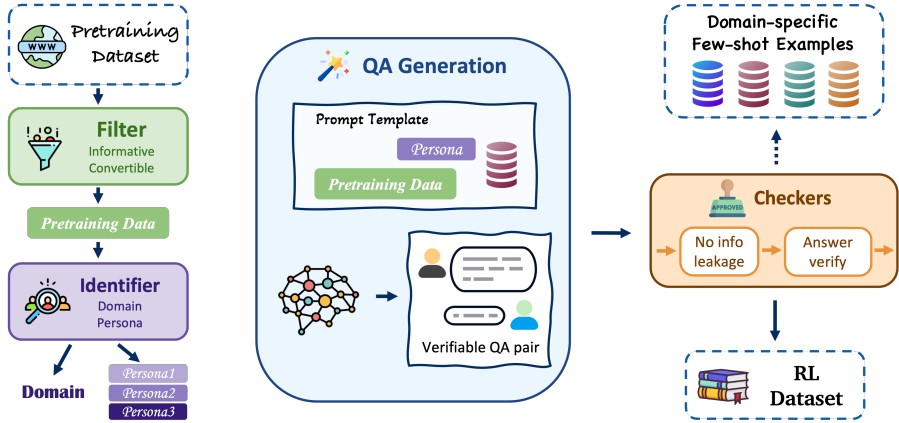

Figure 2: Overview of the `Webscale-RL` data pipeline that systematically converts large-scale pretraining data into RL data while preserving the scale and diversity of web data. The pipeline maintains a domain-specific demonstration library for few-shot examples for high quality generation and assigns multiple personas to each document to encourage reflecting different viewpoints. The generated QA pairs are verified for correctness and leakage prevention to ensure the reliability of the RL dataset. The prompt templates of four stage are listed in Appendix B.1.1.

At a high level, `Webscale-RL` leverages a generative model to convert narrative pretraining documents into *verifiable* QA pairs for RL training. To cover a wider range of topics and question styles, we first maintain a domain-specific demonstration library for few-shot examples to guide the generation process. We further assign multiple *personas* to each document to encourage reflecting different viewpoints. Figure 2 illustrates the pipeline, which consists of four main stages:

**Data Filtering**. Our pipeline takes pretraining corpora spanning multiple domains as input instead of focusing on data in limited domains (Toshniwal et al., 2024; Liu et al., 2025a). This stage aims to remove inputs that are unlikely to yield verifiable high-quality questions. We first use heuristics to filter out obviously low-quality documents ($< 50$ tokens) and then employ an LLM for further fine-grained filtering (Gunasekar et al., 2023; Wettig et al., 2024). Different from previous pipelines that strictly filter data from multiple dimensions (e.g., difficulty (Fan et al., 2025; Ma et al., 2025), format (Zhou et al., 2025), with sophisticated reasoning traces (Yuan et al., 2025), etc.), our filter aims to select data for the following stages while maximally preserving the diversity of the original materials. Therefore, the LLM-based filter only identifies and removes (i) non-informative pages where most contents are boilerplate (e.g. navigation, headers, or footers in website html), and (ii) non-self-contained fragments that lack sufficient context to verify answers. This two-stage filtering ensures that the retained documents are both *informative* and *convertible* into verifiable RL data.

**Domain Classification and Persona Assignment**. After filtering, we then classify each document into a specific domain (e.g., commerce, healthcare, social science, etc.) using a LLM-based classifier. Due to extreme high diversity of the pretraining data, our pipeline adopts different few-shot examples for each domain to ensure that the generated questions are contextually appropriate and verifiable, which is absent in existing pipelines (Fan et al., 2025; Yuan et al., 2025). The domain tags are then used to collect relevant few-shot exemplars in the subsequent QA generation step. Additionally, to further enhance the diversity of the generated QA pairs, we assign multiple *personas*

who will be interested in the content to each document (Ge et al., 2024), which defines the style and perspective from which questions will be generated. For example, a document classified under the "healthcare" domain might be assigned personas such as "medical expert," "patient," or "health journalist." This persona assignment encourages reflecting different viewpoints and information needs in question generation given the same document, thereby capturing more information in the source data and enriching the RL dataset's diversity.

**Verifiable QA Generation**. Conditioned on the source document, domain tag, and chosen persona, the LLM-based QA generator produces verifiable question-answer pairs. Specifically, we first sample few-shot examples from the domain-specific demonstration library, a curated pool covering a range of question types and complexities within each domain to ensure that the generated questions are of high quality. We then incorporate all contexts with a prompt template (in Appendix B.1.1) to guide the LLM-based generator to extract diverse question-answer pairs from the perspective of the assigned persona. For question generation, beyond extracting the questions originally contained in the document (Yue et al., 2024), our generator can also raise new questions that are answerable according to the pretraining data. Since the trained model is not allowed to access the source document during RL, we further instruct the generator to provide necessary contexts to ensure that the question is self-contained. Meanwhile, we only require a relatively short and verifiable ground-truth answer (e.g., a number, a name, or a phrase) grounded by the pretraining materials instead of a long explanation or detailed reasoning steps composed by a strong LLM (Yue et al., 2024; Yuan et al., 2025), which significantly reduces the generation complexity and reliance on the backend LLMs. In other words, our generation is to extract the answer from the document instead of distilling from a powerful LLM. This design choice allows us to leverage more cost-effective LLMs for generation while still producing high-quality, verifiable QA pairs suitable for RL training. We provide a conversion example in Appendix B.2.1.

**Quality Check and Leakage Control**. While the most generated question-answer pairs are of high quality, some may still contain errors or hallucinations. To ensure the reliability of the RL dataset, we leverage an LLM-based verifier to implement a multi-stage checking process (Liu et al., 2024; Prabhakar et al., 2025): 1) *Correctness verification*. Unlike accuracy-based post-processing in previous works (Zhou et al., 2025; Ma et al., 2025), our verification assesses the correctness of the answers by checking if the extracted QA data are grounded by the source document, which is much less biased by the backend LLMs and effectively reduces the wrong reward signals during RL training; 2) *Leakage prevention* ensures that the questions do not reveal answers explicitly (e.g., the ground truth is not trivially embedded in the prompt). The verifier filters out any QA pairs that fail to meet these criteria, ensuring that the final dataset truly tests the model's knowledge or reasoning capabilities rather than its ability to summarize or retrieve information directly from the prompt.

The prompt templates of four stages are listed in Appendix B.1.1. The examples of pretraining to RL conversion are in B.2. We further apply data decontamination by lm-eval-harness (Gao et al., 2024) to remove overlaps with the evaluation. With this pipeline, we can systematically convert large-scale pretraining data into a massive, diverse, and verifiable RL-ready dataset that closely matches the scale and diversity of the original pretraining corpus. This approach effectively addresses the RL data scarcity issue and enables scaling up RL training of LLMs across a wide range of tasks and domains. More discussions are described in Appendix B.1.

## 4 WEBSCALE-RL DATASET

### 4.1 DATASET CONSTRUCTION

We construct `Webscale-RL` dataset by running the data pipeline over a subset (∼1M data in total) of the mixture of pretraining corpora including DCLM (Li et al., 2024), Wikipedia (Foundation), MegaMath (Zhou et al., 2025), Stack-v2 (Lozhkov et al., 2024), etc. The choice of pretraining data here aims to cover diverse domains and mimics previous practice on pretraining (Bakouch et al., 2025). The data selection is flexible and can be adjusted based on the target model and application.

In RL data conversion, we use GPT-4.1-mini for domain classification and final quality check, and GPT-4.1 for data filtering and QA generation. As we mentioned in QA generation stage in Sec. 3.2, our pipeline aims to extract answer grounded by the pretraining document instead of distilling from a strong LLM, which reduces the bias and reliance on the backend LLMs. Therefore, the pipeline

can be extended to other open-source LLMs such as GPT-OSS (Agarwal et al., 2025) and Deepseek series (DeepSeek-AI et al., 2025). For each qualified document, we assign up to 3 personas to generate diverse QA pairs. The final dataset contains ~1.2M QA pairs covering 9+ domains. Note that the dataset can easily be further scaled up to the pretraining level with our `Webscale-RL` pipeline. More details of dataset construction are described in Appendix B.2.

## 4.2 DATASET ANALYSIS

We compare our `Webscale-RL` dataset with other widely used pretraining datasets (RedPajama-v2 (Weber et al., 2024), FineWeb-Edu (Penedo et al., 2024b), DCLM-baseline (Li et al., 2024)), SFT datasets (NaturalReasoning (Yuan et al., 2025), Nemotron (Bercovich et al., 2025)) which include reasoning CoT in the answers, and RL datasets ( DeepScaler (Luo et al., 2025), OpenR1-Math (Hugging Face, 2025), OpenThoughts3 (Guha et al., 2025)) which include a ground-truth answer for each question. The detailed comparison is listed in Table 1.

Table 1: The comparison of various datasets with our `Webscale-RL` dataset. The number of data indicates the number of documents (for pretraining datasets) or the number of QA pairs (for SFT and RL datasets). The **scalability** indicates the potential of scaling up the dataset size: DeepScaler has low scalability because it is collected from competitions and relies on human annotation. Other post-training datasets generate answers by distillation but they collect queries from limited sources, which limits the further scaling. In contrast, both the questions and answers in the `Webscale-RL` dataset are converted from and grounded by the pretraining datasets, which can be easily scaled up to pretraining level.

| Dataset | Type | # of data | Domain | Data Source | Scalability |
|---|---|---|---|---|---|
| RedPajama-v2 | Pretrain | >100B | Multi-domain | Web crawling | / |
| FineWeb-Edu | Pretrain | >3B | Multi-domain | Web crawling | / |
| DCLM-baseline | Pretrain | >3B | Multi-domain | Web crawling | / |
| DeepScaler | RL | 40K | Math | Competition and other math datasets | Low |
| OpenR1-Math | SFT/RL | 220K | Math | Distilled from DeepSeek-R1 | Medium |
| OpenThoughts3 | SFT | 1.2M | Math, Code, Science | Distilled from QwQ-32B | Medium |
| NaturalReasoning | SFT | 1.1M | Multi-domain | Converted from pretrain + distillation | High |
| Nemotron | SFT/RL | 3.9M | Math, Code, Science | Distilled from multiple models | Medium |
| Webscale-RL | RL | 1.2M | Multi-domain | Converted from pretrain | High |

The comparison shows that the pretraining corpora are orders of magnitude larger and span broad domains, whereas existing SFT/RL datasets are significantly smaller and often focus on a few areas (notably math and code), which limits coverage of general knowledge and open-ended reasoning found in web-scale text. The Nemotron dataset includes data in other domains such as general QA and safety, which however only constitutes a small portion of the dataset. It is also worth noting that while some datasets have a relatively large data volume (e.g., OpenThoughts3, Nemotron), they still encounter the challenge of further scaling due to their limited sources of queries. In contrast, our `Webscale-RL` dataset is constructed by converting from the pretraining documents, allowing for easy expansion to pretraining scale.

We also obverse that a large fraction of the SFT/RL data is distilled from other teacher models. This couples dataset quality and ceiling to teacher capability and availability. In contrast, `Webscale-RL` is *grounded in source documents*: the generator does not need to solve the problems during construction; instead, we extract verifiable QA pairs from existing texts, reducing the dependence on strong teachers. Furthermore, because both questions and answers are derived from pretraining documents and verified against the source, `Webscale-RL` can scale naturally with the size of available corpora (i.e., the pretraining scale) while maintaining diversity, unlike human-labeled or fully distilled datasets whose growth is bottlenecked by annotation or query generation.

We list the domain distribution of our dataset in Fig. 3 left. `Webscale-RL` spans 9+ domains inherited from pretraining sources, substantially more diverse than most public post-traininig datasets. While we observe that the STEM-related domains (Math, Science, Code) constitute a significant portion of the dataset, it is also worth noting that the underrepresented domains in existing RL datasets, such as Lifestyle ($> 8.6\%$), Commerce ($> 3.3\%$), etc., are well covered in `Webscale-RL`, which are essential for general-purpose assistants.

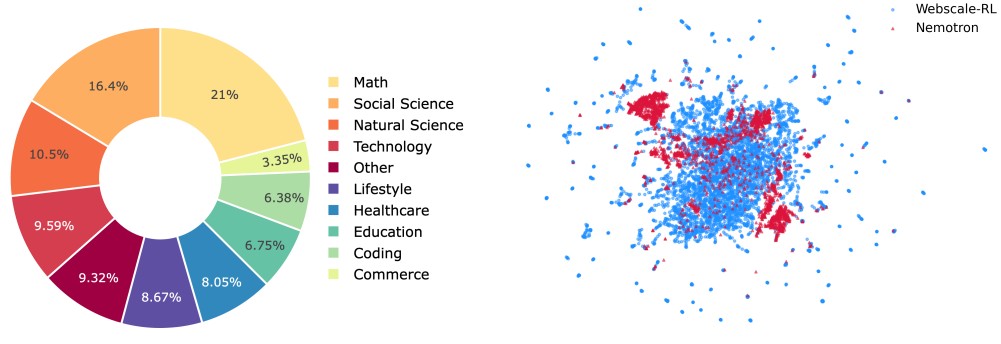

Figure 3: Left: The domain distribution of `Webscale-RL` dataset. Right: The comparison on question embedding of `Webscale-RL` and Nemotron data. We randomly sample 5K questions from each dataset and visualize the embedding (by Qwen3-Embedding) reduced to 2D using UMAP.

To further illustrate the diversity of `Webscale-RL` dataset, we compare it with Nemotron, a large-scale SFT/RL dataset mainly covering math, code, and science. Since we focus on question diversity, we first randomly sample 5K questions from each dataset and encode them using the Qwen3 Embedding model (Zhang et al., 2025b). We then reduce the embedding dimension to 2 using UMAP (McInnes et al., 2018) for visualization. The results are shown in Fig. 3 right. Although both datasets cover multiple domains, Nemotron data points are mainly clustered in several regions, indicating a focus on specific topics. In contrast, the `Webscale-RL` data points are converted from a larger variety of documents and are generated from diverse perspectives by different personas, resulting in a distribution that is more uniform and more scattered, indicating a broader coverage of topics and knowledge areas. The diversity along with the large scale of Webscale-RL can help models learn a wide range of knowledge and reasoning skills, enhancing their versatility and performance across various tasks.

## 5 EXPERIMENTS

In this section, we conduct experiments to evaluate the effectiveness of the `Webscale-RL` dataset generated by our proposed pipeline. Our experiments aim to address two main questions: (1) Can RL data generated by our pipeline enhance model performance across various benchmarks? (2) Does RL training scale more effectively and efficiently than standard teacher-forcing training?

### 5.1 EXPERIMENT SETUP

**Baselines**. To answer these questions, we finetune a Qwen2.5-3B model (Yang et al., 2024a) using GRPO (Shao et al., 2024) on the `Webscale-RL` dataset and compare it with continual pretraining on the corresponding base dataset, i.e., the original pretraining data prior to RL conversion. We further compare our method with several advanced data refinement techniques: (1) QuRating(Wettig et al., 2024), which selects high-quality data via LLM ranking and filtering; (2) ProX(Zhou et al., 2024), which uses programmatic cleaning to enhance data quality; and (3) Generative Data Refinement (GDR) (Jiang et al., 2025), which originally uses LLM to improve the safety of the corpus (e.g., remove personally identifiable information, toxic content). In our experiment, we use it to improve the quality of the pretraining dataset. For these baselines, we refine the pretraining data using each method and perform continual pretraining on the resulting datasets.

Notably, we observe that RL training substantially improves the model's instruction-following abilities, while the continual pretrained models may fail to start answering in the evaluation, especially for questions with zero-shot examples, potentially introducing an evaluation bias. To mitigate this and enable a fair comparison, we construct an SFT dataset comprising 10K high-quality examples. Specifically, we first generate QA pairs via our `Webscale-RL` pipeline and then use GPT-4.1 to distill a relatively short reasoning CoT for each question given the ground-truth answer.

**Training**. For the continual pretraining and data refining baselines, we start from the base model and continue to pretrain on a 1M corpus, which represents a superset of the source data for the `Webscale-RL` dataset. We then follow with SFT training with a smaller learning rate using the 10K high-quality examples. For RL training, we first apply SFT with the same SFT dataset for warm-up. We then sample 150K data points from the `Webscale-RL` dataset and run standard GRPO training. More details of the SFT dataset and training are described in Appendix B.3.

**Benchmarks**. We evaluate the models on a diverse set of benchmarks to assess their general capabilities and domain-specific performance, including general tasks (MMLU-pro (Wang et al., 2024), Big-Bench (Srivastava et al., 2023)), math & STEM tasks (GSM8K (Cobbe et al., 2021), MATH500 (Hendrycks et al., 2021), GPQA-diamond (Rein et al., 2024)) and coding tasks (MBPP (Austin et al., 2021) and EvalPlus (Liu et al., 2023)). For EvalPlus, we report the average score of HumanEval (Chen et al., 2021), MBPP, HumanEval+ and MBPP+. In evaluation, we use the same pipeline and configurations for all models. Specifically, we use zero-shot evaluation for Big-Bench, GPQA-diamond and MATH500. We use 5-shot for MMLU-pro and 8-shot for GSM8K evaluation following the default setting in lm-eval-harness (Gao et al., 2024). More details in evaluation are described in Appendix B.3.

## 5.2 MAIN RESULTS

Table 2 summarizes the comparisons of Webscale-RL with other baselines. Our method outperforms all baselines across most benchmarks, including continual pretraining and advanced data refinement pipelines. We observe an average improvement of 3.4 over the strongest baseline (GDR). Notably, Webscale-RL even narrows the performance gap to the much larger Qwen2.5-7B model from 10.6 pts to 6.1 pts on average. This indicates that converting web-scale corpora into verifiable QA and optimizing with RL yields stronger downstream gains than further imitation on even refined text.

Particularly, the improvements are most pronounced on general knowledge and reasoning tasks (MMLU-pro, Big-Bench, GPQA-diamond), which significantly benefit from the diversity and breadth of the `Webscale-RL` dataset inherited from pretraining sources. On math tasks, we observe a large jump on MATH500 from 47.6 to 58.0 after RL training with Webscale-RL, which is close to the 7B model. This aligns with prior findings that RL can better incentivize math reasoning (Shao et al., 2024; Yang et al., 2024b) compared to simply imitating refined documents or QA demonstrations. The gain on GSM8K is relatively smaller, likely due to the saturation effect as the base model already achieves strong performance. Meanwhile, the performance improvement on coding tasks is relatively smaller, likely reflecting the lower proportion of coding data in the pretraining corpus. Notably, the 3B model finetuned with Webscale-RL substantially narrows the performance gap to the 7B base model on the macro average, suggesting a practical path to stronger small models via RL scaling.

Table 2: Comparison results of our Webscale-RL with baselines on various benchmarks. To mitigate evaluation bias, continual pretraining and data refinement baselines are followed by SFT training to enhance instruction following. While all finetuning are based on the Qwen2.5-3B model, we also compare with the 7B base model. **Blue bold** indicates the best result among 3B baselines; **green bold** shows where we match or exceed the 7B model.

| Method | MMLU-pro | BigBench | GPQA-D | MATH500 | GSM8K | MBPP | EvalPlus | Avg |
|---|---|---|---|---|---|---|---|---|
| Qwen2.5-3B | 37.8 | 41.2 | 20.8 | 47.6 | 74.2 | 54.6 | 57.3 | 47.6 |
| Qwen2.5-7B | 48.3 | 58.7 | 29.6 | 60.8 | 84.4 | 63.4 | 62.2 | 58.2 |
| Cont. Pretrain | 39.9 | 45.1 | 18.6 | 44.0 | 77.4 | **55.2** | **57.8** | 48.3 |
| QuRating | 39.7 | 44.9 | 19.4 | 44.6 | 76.8 | 54.8 | 57.6 | 48.3 |
| ProX | 40.0 | 46.0 | 19.5 | 44.4 | 77.3 | 54.2 | 57.5 | 48.4 |
| GDR | 39.9 | 46.0 | 20.8 | 44.4 | 77.4 | 55.0 | 57.6 | 48.7 |
| **Webscale-RL** | **43.7** | **48.3** | **23.2** | **58.0** | **78.5** | 55.0 | **57.8** | **52.1** |

Despite using a small SFT set to reduce evaluation bias toward instruction-following, RL still maintains clear advantages over SFT-augmented continual pretraining baselines. This suggests that the

gains from Webscale-RL are not solely due to improved instruction adherence but stem from the reward-driven online learning signal. Overall, these results demonstrate that our `Webscale-RL` data pipeline effectively scales up RL data by converting from pretraining corpus and enables significant capability improvements across diverse tasks and domains.

## 5.3 PERFORMANCE COMPARISON OF SCALING TRAINING

While RL shows remarkable advantages over teacher-forcing training in terms of final performance, we further investigate the scaling efficiency of RL training compared to standard pretraining with respect to the amount of training tokens. To this end, we compare the performance of RL training and continual pretraining at different training scales by varying the amount of data sampled from the `Webscale-RL` dataset and the original pretraining corpus, respectively. Notably, we observe that the length of QA pairs in the `Webscale-RL` dataset differs from the length of document in the original pretraining corpus while their source data are the same. Therefore, for a fair comparison on *token efficiency of the original data*, we compute the token number of RL training by the original pretraining corpus used to generate the `Webscale-RL` dataset instead of the `Webscale-RL` dataset itself. For example, if we generate two 300-token QA pairs from a 4000-token pretraining text, then we count the RL training token number as 4000 instead of 600 when training on these two QA pairs. Note that for continual pretraining with different training data volume, we also apply the same SFT training as a follow-up using the 10K high-quality examples to mitigate the evaluation bias.

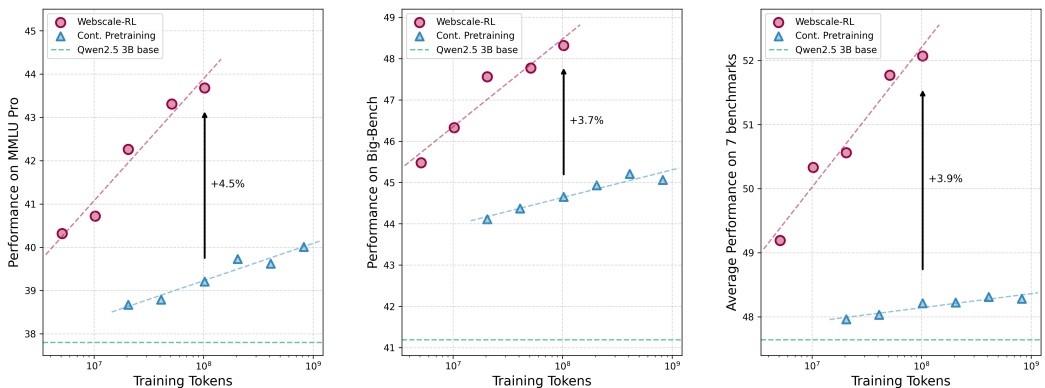

Figure 4: Scaling comparison between Webscale-RL training and continual pretraining with the original pretraining corpora. We report the performances on MMLU-pro (left), Big-Bench (middle) and average on all benchmarks (right). The token number for RL training is calculated based on the original pretraining corpus used to generate the `Webscale-RL` dataset. The each data point in continual pretraining baselines are followed by a SFT training using the same 10K high-quality examples. The RL training on `Webscale-RL` consistently outperforms continual pretraining at different training scales and exhibits better scaling efficiency.

Since the pretraining corpus mainly consists of general web text, we focus on evaluating the models on general tasks (MMLU-pro and Big-Bench) to better reflect the impact of training scale. We also report the average performance across all benchmarks to provide a holistic view.

Figure 4 illustrates the performance comparison between RL training with `Webscale-RL` dataset and continual pretraining with pretraining corpora at different training scales. We observe that RL training consistently outperforms continual pretraining across all three metrics (MMLU-pro, Big-Bench, and average performance) at various training scales. With the same amount of training tokens (100 millions), RL training achieves 4.4% improvement over Qwen2.5-3B base model in average while continual pretraining exhibits similar performance to the base model.

Notably, RL training achieves comparable or better performance with significantly fewer training tokens. For instance, on MMLU-pro, RL training with approximately 10M tokens attains similar performance to continual pretraining with 1B tokens, indicating over $100\times$ improvement in data efficiency. Furthermore, RL training exhibits a steeper upward trend as the training scale increases,

which is also true for other benchmarks, demonstrating that RL training not only leads to higher final performance, but scales more effectively and efficiently than standard teacher-forcing approaches.

## 6 CONCLUSION

In this paper, we introduced the Webscale-RL pipeline, an end-to-end data engine that converts web-scale pretraining corpora into verifiable, RL-ready data while preserving diversity. With this pipeline, we constructed the Webscale-RL dataset, which is orders of magnitude larger and more diverse than existing RL datasets. Empirically, training a LLM with RL on Webscale-RL improves performance across a diverse suite of benchmarks and delivers better data efficiency than continual pretraining at comparable token budgets, especially on general knowledge and open-ended reasoning (MMLU-pro, Big-Bench), with consistent improvements in math and STEM areas.

While our results demonstrate the promise of scaling RL data to pretraining levels, several limitations and future directions remain. The current Webscale-RL dataset lacks coverage of high-quality data in certain domains such as coding, which leads to smaller gains on coding benchmarks. Therefore, one future direction is to rebalance the domain distribution of the pretraining sources according to the target applications (e.g., to integrate repository-scale code data to enhance the coding capability). Meanwhile, the current RL training employs a generative reward model that provides binary feedback based on match with the ground truth. While this reward exhibits high performance and stability for RL training, it introduces a substantial extra inference cost, becoming one bottleneck for scaling up. Future work can explore more efficient reward models to further scale up RL training to larger models and datasets.

## 7 REPRODUCIBILITY STATEMENT

Our Webscale-RL data pipeline is built upon publicly available datasets and publicly available LLMs for generation and verification. The detailed data sources, prompts, and implementation details are described in Appendix B.1 and Appendix B.2. For the continual pretraining and RL finetuning, we use the standard pretraining and RL algorithms (GRPO), and the hyperparameters are detailed in Appendix B.3.

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

## A    USAGE OF LLMS

In paper writing, the LLMs are mainly used for proofreading and polishing the language, including grammar, spelling, and clarity. The main content, ideas, experiments and following presentations (e.g., results and visualizations) are done by the authors. The LLMs assist to draft the results analysis and conclusion sections based on the experimental results provided by the authors. The authors carefully checked the content and made necessary modifications to ensure the accuracy and correctness of the statements.

## B    DETAILS OF DATASET CONSTRUCTION AND TRAINING

### B.1    WEBSCALE-RL DATA PIPELINE DETAILS

Our data pipeline employs GPT-4.1-mini for domain classification and quality checking, while utilizing GPT-4.1 for QA generation to ensure higher quality outputs. In the second stage, we assign up to 3 different personas to each document and generate tailored QA pairs for each persona respectively.

#### B.1.1    PROMPT TEMPLATES

Our pipeline consists of four main stages, each with carefully designed prompts to ensure high-quality data generation:

---

**Stage 1: Data Filtering**

**Role:** Data Analyst
**Objective:** Identify whether the material meets quality criteria for QA generation
**Prompt:**

> *You are a helpful data analyst. You will be given a material which can come from very diverse sources and may not be well-structured. In this stage, your task is to identify whether the material is qualified for the following criteria:*
>
> - *The material is informative and self-contained for the user*
> - *It's possible to extract question and corresponding answer from the material*
> - *The content has sufficient depth and clarity*
>
> *Based on the above instructions, identify whether the material is qualified or not.*
> {Raw Document}

---

**Stage 2: Domain Classification & Persona Assignment**

**Role:** Data Analyst
**Objective:** Classify domain and identify target personas
**Prompt:**

> *You are a helpful data analyst. You will be given a material which can come from very diverse sources and may not be well-structured. In this stage, your task is to identify the domain and persona of the material.*
> *Here are the instructions for the domain and persona:*
>
> - *The domain is the main topic of the material. You should choose from the following domains:* {All Domains}
> - *The persona is the intended audience of the material. If the material is intended for multiple personas, you should list several personas that will be interested in the material*
>
> *Based on the above instructions, identify the domain and persona of the material.*
> {Raw Document}

---

---

**Stage 3: QA Generation**

**Role:** Domain Expert (Persona-specific)
**Objective:** Generate high-quality question-answer pairs from source material
**Prompt:**

> *You will be given a material which can come from very diverse sources and may not be well-structured. In this stage, your task is to generate a question and answer pair from the material.*
>
> *Here are the instructions for the question and answer generation:*
>
> - *You will act as a given persona. You should generate a question and answer pair from your perspective*
> - *Both the question and answer should be totally from the material. Do not generate any information that is not in the material*
> - *You should generate such a question that its corresponding answer is relatively short and can be easily and clearly verified*
> - *Ensure the question is natural and reflects genuine curiosity from the target persona*
>
> {Few-shot Examples}
>
> *Based on the above instructions and examples, generate a question and answer pair from the material.*
>
> {Raw Document}
> {Persona}

---

**Stage 4: Quality Check**

**Role:** Data Labeler
**Objective:** Verify QA pair correctness and detect information leakage
**Prompt:**

> *You are a data labeler. You will be given a material and a question and answer pair generated from the material. Your task is to check whether the question and answer pair is correct according to the material and whether there is info leakage from question to answer.*
>
> *Here are the instructions for checking:*
>
> - *For the answer correctness, you should check whether the answer is correct according to the original material*
> - *The information leakage indicates that the question explicitly provides information about the answer and then the answer can be directly obtained from the question*
> - *Ensure the question requires genuine understanding of the source material*
>
> {Few-shot Examples}
>
> *Based on the above instructions, check the QA pair extracted from the original material in terms of the answer correctness and info leakage.*
>
> {Raw Document}
> {QA Pair}

## B.2 WEBSCALE-RL DATASET COMPOSITION

We curate our dataset from diverse pretraining corpora to ensure comprehensive domain coverage while emphasizing reasoning capabilities. The selected sources include DCLM (Li et al., 2024), Wikipedia (Foundation), MegaMath (Zhou et al., 2025), Stack-v2 (Lozhkov et al., 2024), with additional data from OpenMathReasoning (Moshkov et al., 2025) and OpenCodeReasoning (Ahmad et al., 2025) following SmolLM3 (Bakouch et al., 2025) protocols.

Table 3: Source distribution of the Webscale-RL dataset (∼1.2M total QA pairs)

| Source Dataset | # of Converted QA Pairs |
|---|---|
| DCLM | ∼550K |
| Wikipedia | ∼350K |
| MegaMath | ∼100K |
| OpenMathReasoning | ∼100K |
| Stack-v2 | ∼50K |
| OpenCodeReasoning | ∼50K |

### B.2.1 DATA CONVERSION EXAMPLE

The following example demonstrates our persona-driven conversion process:

---

**Original Wikipedia Document: Alterna Bank**

CS Alterna Bank (), operating as Alterna Bank (), is a Canadian direct bank and a wholly owned subsidiary of the Ontario-based credit union Alterna Savings. The bank offers chequing and high-interest savings accounts and mortgages.

Operating primarily as a direct bank since 2017, most customers access accounts using the bank's website, telephone service, and mobile apps. Unlike most other direct banks, some accounts can also be accessed through branches. There are two Alterna Bank locations in Gatineau, QC, and Alterna Savings branches also administer deposits and loans on its behalf...

The bank originated as the Civil Service Loan Corporation, founded 29 October 1992 and operating as CS Loan Corporation. It became CS Alterna Bank after receiving letters patent of continuation on 2 October 2000 as a federally regulated institution under the Bank Act...

Alterna Bank is a member of Canada Deposit Insurance Corporation (CDIC)...

---

**Converted QA Pair: Financial Analyst Persona**

**Question:** In examining the regulatory protection for depositors, is Alterna Bank a member of the Canada Deposit Insurance Corporation (CDIC)?
**Answer:** Yes, Alterna Bank is a member of Canada Deposit Insurance Corporation (CDIC).

---

**Converted QA Pair: Commerce Student Persona**

**Question:** In Canadian direct banking, what is notable about the way Alterna Bank allows its customers to access their accounts compared to most other direct banks?
**Answer:** Some Alterna Bank accounts can be accessed through branches, unlike most other direct banks.

---

### B.3 TRAINING IMPLEMENTATION DETAILS

### B.3.1 BASELINE IMPLEMENTATION

**Generative Refinement Baseline:** Following (Jiang et al., 2025), we adapt their safety-focused approach to quality improvement. GPT-4.1 processes each document by: (1) assessing content quality similar to our filtering stage, returning original text if adequate; (2) refining documents by removing non-informative sections or discarding low-quality content entirely.

**SFT Dataset Construction:** Our 10K SFT dataset enhances instruction-following capabilities post-continual pretraining and provides RL training warmup. We sample 10K queries from a held-out Webscale-RL subset with no training overlap. Since original answers are concise, GPT-4.1 generates detailed Chain-of-Thought explanations based on ground truth, reducing hallucination compared to full model distillation.

### B.3.2 TRAINING HYPERPARAMETERS

Table 4: Training configuration and hyperparameters

| Training Stage | Hyperparameter | Value |
|---|---|---|
| **Continual Pretraining** | Batch Size | 256 |
| | Learning Rate | $1 \times 10^{-5}$ |
| | Max Input Length | 4096 |
| **Supervised Fine-tuning** | Batch Size | 128 |
| | Learning Rate | $5 \times 10^{-6}$ |
| | Max Input Length | 4096 |
| **Reinforcement Learning** | Batch Size | 256 |
| | Learning Rate | $5 \times 10^{-6}$ |
| | Samples per Query | 16 |
| | Max Rollout Length | 2560 |
| | Algorithm | GRPO (Shao et al., 2024) |

All experiments use AdamW optimizer with VeRL (Sheng et al., 2025) as the training backend. RL training employs binary rewards, where an LLM judges whether generated answers match ground truth responses.

### B.3.3 EVALUATION FRAMEWORK

Table 5: Evaluation benchmarks and configurations

| Benchmark | Framework | Shots | Domain Focus |
|---|---|---|---|
| MMLU-Pro | LM-Eval | 5 | Multi-domain Knowledge |
| BigBench | LM-Eval | 0 | Reasoning & Language |
| GPQA-D | LightEval | 0 | Scientific Reasoning |
| MATH500 | LightEval | 0 | Mathematical Problem Solving |
| GSM8K | LM-Eval | 8 | Grade School Math |
| MBPP | EvalPlus | 0 | Python Programming |
| EvalPlus | EvalPlus | 0 | Code Generation & Testing |

We employ LM-eval-harness (Gao et al., 2024), LightEval (Habib et al., 2023), and EvalPlus (Liu et al., 2023) with default settings for prompt templates, metrics, and decoding parameters. MMLU-Pro and GSM8K use few-shot evaluation (5 and 8 shots respectively) following standard protocols, while other benchmarks use zero-shot evaluation.

