# OpenReview forum: "Webscale-RL: Automated Data Pipeline for Scaling RL Data to Pretraining Levels"
_ICLR.cc/2026/Conference — ICLR 2026 Poster_

### Official Review · Reviewer_cyfs · 2025-10-27

**Soundness:** 3
**Presentation:** 3
**Contribution:** 4
**Rating:** 8
**Confidence:** 4

**Summary:**

The paper introduces Webscale-RL, an automated and scalable data pipeline designed to bridge the data bottleneck in reinforcement learning (RL) for large language models (LLMs). The core idea is to systematically convert large-scale pretraining corpora into verifiable question–answer (QA) pairs, enabling RL training at web scale. Using this pipeline, the authors construct the Webscale-RL dataset, consisting of 1.2 million QA pairs across nine or more domains. Empirical results demonstrate that models trained with RL on this dataset outperform continual pretraining and advanced data refinement baselines across a wide range of benchmarks (MMLU-Pro, BigBench, MATH500, etc.), achieving comparable or superior performance with up to 100× fewer tokens. Overall, the work aims to make RL training as scalable as pretraining, thus improving both capability and efficiency of LLMs.

**Strengths:**

1. The paper addresses one of the central challenges in applying RL to LLMs — the lack of large, diverse, and verifiable datasets. The proposed Webscale-RL pipeline is a substantial step toward scaling RL data generation to the same order of magnitude as pretraining corpora, a problem of clear importance for the community.

2. The data engine is well structured, with clearly defined stages — data filtering, domain classification with persona assignment, QA generation, and multi-stage verification. This modularity enhances reproducibility and potential adaptability to new domains.

3. The proposed framework could substantially reduce human labeling costs and enable scalable RL training for future LLM development. The paper provides sufficient implementation detail for reproducibility.

4. Experiments are extensive and convincing. RL training with Webscale-RL yields consistent gains across general reasoning, math, and STEM benchmarks. The results on data efficiency (100× fewer tokens to reach equivalent performance) are particularly impressive and suggest genuine benefits beyond data refinement or imitation-based methods.

**Weaknesses:**

1. Although multi-domain, the dataset underrepresents certain crucial areas (e.g., coding, tool use). This imbalance is reflected in weaker improvements on programming benchmarks (e.g., MBPP), limiting generalizability across task types.

2. The pipeline relies heavily on GPT-4.1 models for data filtering, classification, generation, and verification. This dependency may restrict reproducibility for groups without access to such models and raises cost and sustainability concerns.

3. Experiments are performed mainly on a 3B-parameter model (Qwen2.5-3B). While the improvements are substantial, it remains unclear how well the approach scales to larger or smaller models, or to different RL algorithms. Real-world downstream evaluations (dialogue, reasoning under uncertainty, tool use) are also absent.

**Questions:**

none

---

> ### Author Response · Authors · 2025-11-26
> **response to reviewer cyfs (1/2)**
>
> We thank the reviewer for your thoughtful and constructive feedback. Here are our responses:
> - **W1. Although multi-domain, the dataset underrepresents certain crucial areas (e.g., coding, tool use). This imbalance is reflected in weaker improvements on programming benchmarks (e.g., MBPP), limiting generalizability across task types.**
>
>   Thanks for pointing it out. In our experiment, the recipe of source data construction mimics the existing pretraining dataset, which has a relatively low proportion in some domains such as coding. We also agree that it is crucial for performance improvement on these areas to augment more coding data but we leave it a future work.
>
> - **W2. The pipeline relies heavily on GPT-4.1 models for data filtering, classification, generation, and verification. This dependency may restrict reproducibility for groups without access to such models and raises cost and sustainability concerns.**
>
>   Thank you for this insightful question. Although we use GTP-4.1 / 4.1-mini in data conversion, our data pipeline aims to extract questions and answers from the pretraining materials instead of solving the questions by the backend LLM. In verification step, the LLMs check whether the answer is grounded by the original data instead of judging based on their own knowledge. Therefore, in practice, our data pipeline depends less on the powerful frontier models compared to existing distillation-based pipelines that construct datasets by distilling the answers from frontier LLMs.
>
>   Take the "generation" step (the most important stage in our pipeline) for example, we also provide a comparison how different LLMs influence the generated data.  We use the "Sound speed profile" from wikipedia pretraining dataset as the source data (see full text in its [wiki page](https://en.wikipedia.org/wiki/Sound_speed_profile)) and set the persona as "Acoustic researchers". Here are the generated QA pairs from different LLMs.
>   - Ours (GPT4.1 and 4.1-mini mixed).
>     ```
>     Question: In underwater acoustics research, understanding which variables most significantly affect the speed of sound at a given ocean depth is crucial for modeling and measurements. What are the main factors that determine the speed of sound at a certain depth in the ocean, in order of their importance?
>     Answer: The speed of sound at a given ocean depth depends on the temperature at that depth, the depth itself, and the salinity at that depth, in that order.
>     ```
>   - GPT-oss-120B.
>     ```
>     Question: In ocean acoustics, a common feature of sound speed profiles is a channel whose axis lies at the depth of minimum sound speed, allowing sounds emitted near this axis to travel long horizontal distances. What is this channel called?
>     Answer: The SOFAR channel.
>     ```
>   - Llama-3.3-70B-instruct.
>     ```
>     Question: In the ocean, what are the primary factors that affect the speed of sound at a given depth, and in what order of importance?
>     Answer: The speed of sound depends on the temperature at that depth, the depth itself, and the salinity at that depth, in that order.
>     ```
>   - Qwen3-32B.
>     ```
>     question: What are the two general representations of a sound speed profile?
>     answer: The two general representations are tabular form (with pairs of columns for depth and sound speed) and a plot (with depth on the vertical axis and sound speed on the horizontal axis).
>     ```
>
>   It show that the weaker LLMs can also extract correct QA pairs. In practice, if people have concerns on cost or privacy issues, they can also use other open-source models (e.g., gpt-oss or deepseek series) as backend models in our pipeline.

---

> > ### Author Response · Authors · 2025-11-26
> > **response to reviewer cyfs (2/2)**
> >
> > - **W3. it remains unclear how well the approach scales to larger or smaller models, or to different RL algorithms. Real-world downstream evaluations (dialogue, reasoning under uncertainty, tool use) are also absent.**
> >
> >   Due to the limited computation resources, we didn't scale up to larger models with similar scale or other RL algorithms. The main objective of our experiment is to show that compared to teacher-foring learning (pretraining), converting pretraining data to RL data and RL training can be more efficient paradigm to utilize the same data. Since we use the standard GRPO for experiment, we believe it can be extended to other more advanced RL algorithms.
> >
> >   For downstream evaluation, since the tool-use data is seldom covered by the pretraining data, we focus on the reasoning tasks. We run RL on DAPO-17K dataset with standard GRPO and compare Webscale-RL and continual pretraining. We evaluate their performances on challenging math competition benchmarks. We report pass@32 for AIME and pass@1 for others.
> >   |    | Minerva | Olympiad | AMC  | AIME24 | AIME25 |
> >   | -- | ------- | -------- | ---- | ------ | ------ |
> >   | cont. pretrain + DAPO-17K RL | 29.4    | 26.2     | 35.0 | 5.5    | 2.7    |
> >   | Webscale-RL + DAPO-17K RL     | 32.0    | 28.3     | 42.5 | 7.5    | 3.9    |
> >
> >   The results shows that Webscale-RL can still exhibits advantages in downstream domain-specific training.
> >
> > We sincerely appreciate your effort again and we hope our responses can address your concerns.

---

### Official Review · Reviewer_QNn9 · 2025-11-01

**Soundness:** 3
**Presentation:** 3
**Contribution:** 3
**Rating:** 4
**Confidence:** 3

**Summary:**

The paper introduces Webscale-RL pipeline, a scalable data engine that systematically converts large-scale pre-training documents into millions of diverse, verifiable question-answer pairs for RL. Using this pipeline, the paper constructs the Webscale-RL dataset, containing 1.2 million examples across more than 9 domains.

**Strengths:**

1. The construction pipeline of this paper is solid and the proposed method is sound.
2. The manuscript is also well-written and easy to follow.

**Weaknesses:**

1. The motivation and methodology of this paper are quite similar to NaturalReasoning (Yuan et al., 2025), which significantly limits the novelty of this work.

2. The entire data generation and annotation pipeline heavily relies on GPT-4.1 and GPT-4.1-mini. This raises concerns about potential biases introduced by these models, and whether the associated costs are acceptable, especially when compared to the scale and cost-efficiency of large-scale pretraining corpora.

3. The experiments and analysis in this paper are quite limited — almost no in-depth analysis is provided. At the very least, a comparative experiment with NaturalReasoning should be included. Moreover, using RL as a post-training method and comparing it against continued pretraining may not be a fair comparison.

**Questions:**

See above.

---

> ### Author Response · Authors · 2025-11-26
> **response to reviewer QNn9 (1/2)**
>
> We thank the reviewer for your thoughtful and constructive feedback. Here are our responses:
>
> - **W1. The motivation and methodology of this paper are quite similar to NaturalReasoning, which significantly limits the novelty of this work.**
>
>   We appreciate the reviewer highlighting the connection to NaturalReasoning. While both works extract QA pairs from pretraining corpora, our motivation and problem setting differ from it.
>
>   For motivation, NaturalReasoning aims to build a large-scale synthetic SFT corpus by generating long CoT answers through model distillation. In contrast, our goal is to construct a large-scale RL dataset, where both the question and the answer are directly grounded in the original pretraining materials instead of model-generated reasoning.
>
>   The difference in motivation also leads to different pipeline design: NaturalReasoning mainly relies on a strong model to compose answer so its pipeline focuses mainly on filtering and question extraction. Our pipeline incorporates additional steps such as persona assignment and quality check, ensuring that answers remain faithful to the source document without adding hallucinated reasoning. Furthermore, NaturalReasoning focuses on large scale SFT whereas we studies large-scale RL training, including final performance and token efficiency.
>
>   Therefore, while we acknowledge conceptual overlap in using pretraining corpora as a data source, we believe our motivation, pipeline design, and experimental scope introduce meaningful novelty.
>
> - **W2. The entire data generation and annotation pipeline heavily relies on GPT-4.1 and GPT-4.1-mini. This raises concerns about potential biases introduced by these models, and whether the associated costs are acceptable, especially when compared to the scale and cost-efficiency of large-scale pretraining corpora.**
>
>   Thank you for this insightful question. A key clarification is that our pipeline does not ask the LLM to solve a question. Instead, it asks the model to extract a question–answer pair whose correctness is strictly grounded by the source document. Since answers must be verbatim or directly inferable from the pretraining text, the LLM’s own knowledge plays a limited role compared to distillation-style synthetic datasets. Therefore, our approach substantially reduces the dependence on model knowledge and thereby mitigates bias.
>
>   To further illustrate it, we provide an example for the generation comparison of different LLMs. We compared generated QA pairs from several models using the same pretraining document (“Sound speed profile” from [wiki page](https://en.wikipedia.org/wiki/Sound_speed_profile)) with an identical persona (“Acoustic researchers”). Here are the generated QA pairs from different LLMs.
>   - Ours (GPT4.1 and 4.1-mini mixed).
>     ```
>     Question: In underwater acoustics research, understanding which variables most significantly affect the speed of sound at a given ocean depth is crucial for modeling and measurements. What are the main factors that determine the speed of sound at a certain depth in the ocean, in order of their importance?
>     Answer: The speed of sound at a given ocean depth depends on the temperature at that depth, the depth itself, and the salinity at that depth, in that order.
>     ```
>   - GPT-oss-120B.
>     ```
>     Question: In ocean acoustics, a common feature of sound speed profiles is a channel whose axis lies at the depth of minimum sound speed, allowing sounds emitted near this axis to travel long horizontal distances. What is this channel called?
>     Answer: The SOFAR channel.
>     ```
>   - Llama-3.3-70B-instruct.
>     ```
>     Question: In the ocean, what are the primary factors that affect the speed of sound at a given depth, and in what order of importance?
>     Answer: The speed of sound depends on the temperature at that depth, the depth itself, and the salinity at that depth, in that order.
>     ```
>   - Qwen3-32B.
>     ```
>     question: What are the two general representations of a sound speed profile?
>     answer: The two general representations are tabular form (with pairs of columns for depth and sound speed) and a plot (with depth on the vertical axis and sound speed on the horizontal axis).
>     ```
>
>   Across models, we consistently observe that the generated QA pairs are faithfully grounded in the provided source text, underscoring that the required capability is document comprehension, not advanced reasoning. This suggests that the pipeline is compatible with capable open-source models (e.g., GPT-OSS, DeepSeek-series), offering flexibility depending on cost or privacy constraints.
>
>   Thus, although we used GPT-4.1/mini in our experiment, the risk of bias is limited and the pipeline remains broadly applicable and cost-controllable.

---

> ### Author Response · Authors · 2025-11-26
> **response to reviewer QNn9 (2/2)**
>
> - **W3a. The experiments and analysis in this paper are quite limited ... A comparative experiment with NaturalReasoning should be included.**
>
>   We appreciate the reviewer’s suggestions and conducted additional experiments to strengthen the comparison with NaturalReasoning.
>
>   To compare with NaturalReasoning fairly, we first filter the NaturalReasoning dataset and only keep the data with a verifiable answer. Since the NaturalReasoning only uses DCLM and FineMath as the source data, we match it by selecting only the DCLM-derived and math-domain data from our Webscale-RL dataset. We then use the same starting model (SFTed Qwen2.5-3B) to run RL for 200 steps (~100K data) on NaturalReasoning and our dataset separately. The evaluation result is shown as follows:
>   |    | MMLU-pro | Big-Bench | GPQA-D   | MATH500  | GSM8K    |
>   | -- | -------- | --------- | -------- | -------- | -------- |
>   | Qwen2.5-3B       | 37.8     | 41.2      | 20.8     | 47.6     | 74.2     |
>   | NaturalReasoning | 42.7     | 46.8      | **24.3** | 55.6     | 76.7     |
>   | Webscale-RL (DCLM + math domain)     | **43.2** | **48.0**  | 23.4     | **58.8** | **78.1** |
>
>   Our dataset shows higher performance on MMLU-pro, Big-Bench and math benchmarks while NaturalReasoning performs better on GPQA-diamond. We believe this highlights complementary strengths and validates the value of our grounding-based RL data.
>
> - **W3b. Using RL as a post-training method and comparing it against continued pretraining may not be a fair comparison.**
>
>   We agree that RL and continual pretraining differ in nature. Our goal is not to claim that one paradigm dominates the other, but to explore whether a grounded RL dataset can serve as a scalable and token-efficient alternative for utilizing pretraining corpora.
>
>   Meanwhile, we also try to make the comparison fair by multiple mitigations including: 1) we apply SFT on both RL and continual pretraining with the same data to align their capabilities on instruction following; and 2) when varying the token budget, we count the used tokens according to the pretraining data for RL training (see line 440~446 in paper). While some differences are unavoidable, we believe this comparison provides useful insight and motivates future work on scalable RL-based training.
>
> We sincerely appreciate your effort again and we hope our responses can address your concerns. Feel free to let us know if further clarification would be helpful.

---

### Official Review · Reviewer_giqw · 2025-11-01

**Soundness:** 3
**Presentation:** 2
**Contribution:** 3
**Rating:** 6
**Confidence:** 3

**Summary:**

This study presents an automated pipeline for scaling reinforcement-learning (RL) data generation to pretraining-level diversity and scale, and it  identifies a fundamental bottleneck: RL training data (<10 B tokens) are vastly smaller and narrower than web-scale pretraining corpora (>1 T tokens). To bridge this gap, the authors propose a scalable RL-data engine that converts pretraining documents into verifiable question–answer pairs; and provide a webscale-RL dataset, containing ≈ 1.2 M verified QA pairs across 9 + domains (math, code, science, lifestyle, commerce, etc.), derived directly from pretraining corpora and easily extendable to trillions of tokens;  Empirical evidence that RL training on Webscale-RL yields large gains over continual pretraining and data-refinement baselines.

**Strengths:**

- Novel scalable data pipeline: first systematic approach to transform general pretraining corpora into verifiable RL data; elegant combination of domain and persona-based generation with automated quality control.

- Massive coverage & diversity: dataset spans 9 + domains, addressing the narrow-domain limitation of prior RL datasets (mostly math / code).

- Practical contribution: reusable, reproducible pipeline that could standardize RL-data construction for open-source models.

**Weaknesses:**

- Limited transparency of generation details: prompt templates, filtering thresholds, and judge-LLM configurations are summarized only in appendices; reproducibility may depend on access to proprietary models (GPT-4.1).


- Evaluation bias: The SFT “warm-up” and differing token budgets between RL and continual pretraining make it difficult to ensure the strict fairness of comparison.


- Domain imbalance: coding and certain STEM domains remain underrepresented, reflected in smaller gains on code benchmarks.


- The cost and environmental impact of multi-stage LLM generation and verification are not analyzed.

**Questions:**

- How robust are results if GPT-4.1 is replaced by a smaller verifier? Does verification bias the dataset toward specific model families?


- Have you tested pipeline scalability beyond 1.2 M pairs, e.g., to 100 M, regarding quality degradation, cost, or RL performance saturation?


- Since persona assignment drives diversity, do personas from one domain (e.g., healthcare) transfer to unseen domains?

---

> ### Author Response · Authors · 2025-11-26
> **response to reviewer giqw (1/2)**
>
> We thank the reviewer for your thoughtful and constructive feedback. Here are our responses:
> - **W1. Limited transparency of generation details ... reproducibility may depend on access to proprietary models (GPT-4.1).**
>
>   Thank you for pointing out this and we greatly appreciate your suggestions. We re-organize the writing of data pipeline part in the main text and appendix to make it more clear for audiences. The full update can be found in the revision. Here are the summary of our update:
>   - We provide more details of our data pipelines in sec.3.2 and compare it with some existing works;
>   - We highlight the reference of prompt template and conversion example to appendix.
>   - We highlight the backend LLM configuration in main text and add more discussions.
>
>   For the reproducibility and access to GPT-4.1, see Q1 for more discussions.
>
> - **W2. Evaluation bias: The SFT “warm-up” and differing token budgets between RL and continual pretraining make it difficult to ensure the strict fairness of comparison.**
>   - For SFT "warm-up", we clarify that the "warm-up" data is used **both in RL training and continual pretraining** as mentioned in sec.5.1-training. For continual pretraining experiment, we apply SFT on the "warm-up" dataset after continual pretraining on pretraining data to reduce the evaluation bias due to difference in instruction following capbility.
>   - For token budgets. The main motivation of experiment with different token budgets is to compare the training data efficiency given the same source data. Therefore, when computing the token utilization, we count it based on the source pretraining data for both continual pretraining and RL training. i.e., 1) for continual pretraining, we count it by the exactly used pretraining text; 2) for RL training, the RL data are converted from pretraining text, we count it by **the original pretraining text** instead of the RL data. In practice, the average token number of pretraining text is $>4000$ while the converted RL QA pair has $<300$ token in average. We also acknowledge that the training model will rollout new tokens during RL but here we want to highlight the utilization efficiency on the training data instead of training overhead.
>
>   While we understand that there may still be some unavoidable discrepancies in experiment due to the difference in nature of RL and continual pretraining, we apply multiple mitigations (e.g., the same warm-up and token computaion) and we believe we try to compare them in a fair setting to the best of our knowledge.
>
> - **W3. Coding and certain domains remain underrepresented, reflected in smaller gains on code benchmarks.**
>
>   Thanks for pointing it out. In our experiment, we construct the source data by combining the public pretraining dataset, which usually has a relatively low proportion of coding tasks. We also agree that it is crucial for coding performance improvement to augment more coding data but we leave it a future work.
>
> - **W4. The cost and environmental impact of multi-stage LLM generation and verification are not analyzed.**
>
>   With our current configuration (GPT-4.1 and 4.1-mini mixed), the cost of generating 1000 QA pairs is around $8. In practice, our pipeline can also use other open-source models as backend because it depends less on the powerfulness of the backend LLM. See discussion in Q1 for more details.

---

> > ### Author Response · Authors · 2025-11-26
> > **response to reviewer giqw (2/2)**
> >
> > - **Q1. How robust are results if GPT-4.1 is replaced by a smaller verifier? Does verification bias the dataset toward specific model families?**
> >
> >   That's a great question. In our data pipeline, we use LLMs to extract both question and answer from the data instead of asking the LLMs to solve the question themselves. In verification stage, the LLMs check the correctness of answer by checking if it **is grounded by the original pretraining data** rather than judging based on its own knowledge. Therefore, the bias towards the backend LLM in our dataset is much smaller than many existing datasets which are constructed by distilling from backend LLMs.
> >
> >   To further illustrate it, we provide an example for the generation comparison of different LLMs. We use the "Sound speed profile" from wikipedia pretraining dataset as the source data (see full pretraining text in its [wiki page](https://en.wikipedia.org/wiki/Sound_speed_profile)) and set the persona as "Acoustic researchers". Here are the generated QA pairs from different LLMs.
> >   - Ours (GPT4.1 and 4.1-mini mixed).
> >     ```
> >     Question: In underwater acoustics research, understanding which variables most significantly affect the speed of sound at a given ocean depth is crucial for modeling and measurements. What are the main factors that determine the speed of sound at a certain depth in the ocean, in order of their importance?
> >     Answer: The speed of sound at a given ocean depth depends on the temperature at that depth, the depth itself, and the salinity at that depth, in that order.
> >     ```
> >   - GPT-oss-120B.
> >     ```
> >     Question: In ocean acoustics, a common feature of sound speed profiles is a channel whose axis lies at the depth of minimum sound speed, allowing sounds emitted near this axis to travel long horizontal distances. What is this channel called?
> >     Answer: The SOFAR channel.
> >     ```
> >   - Llama-3.3-70B-instruct.
> >     ```
> >     Question: In the ocean, what are the primary factors that affect the speed of sound at a given depth, and in what order of importance?
> >     Answer: The speed of sound depends on the temperature at that depth, the depth itself, and the salinity at that depth, in that order.
> >     ```
> >   - Qwen3-32B.
> >     ```
> >     question: What are the two general representations of a sound speed profile?
> >     answer: The two general representations are tabular form (with pairs of columns for depth and sound speed) and a plot (with depth on the vertical axis and sound speed on the horizontal axis).
> >     ```
> >
> >   We can observe that the listed models are capable of extracting QA pairs that are grounded by the original pretraining data. Therefore, although we used GPT-4.1/mini, the risk of bias is limited and the pipeline remains broadly applicable and cost-controllable by using other capable open-source models.
> >
> > - **Q2. Have you tested pipeline scalability beyond 1.2 M pairs, e.g., to 100 M, regarding quality degradation, cost, or RL performance saturation?**
> >
> >   We did not test the scalability to a much larger scale for either data generation or RL training. For the data generation, our data pipeline can be well scaled to a much larger scale by inputing more pretraining data. To reduce the cost, we can also use open-source models (e.g., gpt-oss as mentioned above or deepseek, etc.) as backend. For the RL performance, due to the limited computation resources, we are not able to run RL training in a larger scale but other works [1] observe that there is a saturation when scaling up RL.
> >
> > - **Q3. Since persona assignment drives diversity, do personas from one domain (e.g., healthcare) transfer to unseen domains?**
> >
> >   In our pipeline, the persona is assigned according to the corresponding pretraining material. It's possible to transfer a persona in healthcare domain (e.g., patient) to a new data if it covers the topic to the interest of a patient. However, in practice, we will re-assign persona for each pretraining data.
> >
> >
> > We sincerely appreciate your effort again and we hope our responses can address your concerns.
> >
> > [1] Khatri, Devvrit, et al. "The art of scaling reinforcement learning compute for llms." arXiv preprint arXiv:2510.13786 (2025).

---

### Official Review · Reviewer_TSDn · 2025-11-03

**Soundness:** 3
**Presentation:** 3
**Contribution:** 3
**Rating:** 6
**Confidence:** 3

**Summary:**

This paper studies the scaling of reinforcement learning dataset. They propose to utilize LLM to convert pretraining dataset into verifiable QA pairs. The empirical evaluation shows that this is much more efficient than naive continual pretraining, providing a potential way to scaling rl and the possibility to change the pretraining/rl paradigm.

**Strengths:**

1. This paper demonstrates the viability of converting pretraining data into reinforcement data. This is a rather novel research direction compared to ones that simply do filtering or rephrasing of pretraining data.

**Weaknesses:**

1. There seems to be a limitation of diversity for generating verifiable answers. The correctness also might depend on the powerfulness of the LLM used for conversion.
2. The description of the proposed method seems to be a bit overly succinct. Even though some parts of the pipelines might share similarities with the paper cited. For completeness, it is better to expand the description in Sec 3.2 and clearly discuss the difference with the prior works. The section for verifiable generation seems to be especially important for further discussions.

**Questions:**

1. Could the authors expand the descriptions of Sec 3.2 to specify more details of each pipeline stage?
2. Do the authors have any ablation to show the diversity of the generated rl data and how it correlates with the final performance (Are there any redundancy in the generation thus the performance seems to be saturating or are there  room for further improvements).

---

> ### Author Response · Authors · 2025-11-26
> **response to reviewer TSDn (1/2)**
>
> We thank the reviewer for your thoughtful and constructive feedback. Here are our responses:
> - **W1. There seems to be a limitation of diversity for generating verifiable answers. The correctness also might depend on the powerfulness of the LLM used for conversion.**
>
>   For the diversity, our data pipeline constructs Webscale-RL dataset by converting pretraining data to RL data while preserving the diversity of the original pretraining corpora. As a result, compared to existing datasets (in table 1), the webscale-rl dataset shows higher diversity in terms of domain amount and semantic embedding as shown in fig.3. We believe the diversity is one major strengh of our approach on scaling verifiable RL data.
>
>   For the correctness of answer generation, a key clarification is that our pipeline does not ask the LLM to solve a question. Instead, it asks the model to extract a question–answer pair whose correctness is strictly grounded by the source document, which substantially reduces the dependence on model knowledge. Therefore, we argue that our method relies less on the powerfulness of the LLM especially when compared to many RL datasets generated by distillation.
>
>   Here is an example for the generation comparison of different LLMs. We compare generated QA pairs from several models using the same pretraining data (“Sound speed profile” from [wiki page](https://en.wikipedia.org/wiki/Sound_speed_profile)) with an identical persona (“Acoustic researchers”). Here are the generated QA pairs from different LLMs.
>   - Ours (GPT4.1 and 4.1-mini mixed).
>     ```
>     Question: In underwater acoustics research, understanding which variables most significantly affect the speed of sound at a given ocean depth is crucial for modeling and measurements. What are the main factors that determine the speed of sound at a certain depth in the ocean, in order of their importance?
>     Answer: The speed of sound at a given ocean depth depends on the temperature at that depth, the depth itself, and the salinity at that depth, in that order.
>     ```
>   - GPT-oss-120B.
>     ```
>     Question: In ocean acoustics, a common feature of sound speed profiles is a channel whose axis lies at the depth of minimum sound speed, allowing sounds emitted near this axis to travel long horizontal distances. What is this channel called?
>     Answer: The SOFAR channel.
>     ```
>   - Llama-3.3-70B-instruct.
>     ```
>     Question: In the ocean, what are the primary factors that affect the speed of sound at a given depth, and in what order of importance?
>     Answer: The speed of sound depends on the temperature at that depth, the depth itself, and the salinity at that depth, in that order.
>     ```
>   - Qwen3-32B.
>     ```
>     question: What are the two general representations of a sound speed profile?
>     answer: The two general representations are tabular form (with pairs of columns for depth and sound speed) and a plot (with depth on the vertical axis and sound speed on the horizontal axis).
>     ```
>
>   We can observe that all of the listed models are capable of extracting correct QA pairs (grounded by the original pretraining data), which further shows that our pipeline does not heavily rely on the powerfulness of LLMs.
>
> - **W2 & Q1. The description is a bit overly succinct ... it is better to expand the description in Sec 3.2 and clearly discuss the difference with the prior works. Expand the descriptions of Sec 3.2 to specify more details of each pipeline stage.**
>
>   We greatly appreciate your valuable suggestions. Now we provide more details of each stage in generation pipeline and compare our pipeline with some existing methods as suggested. The full update can be found in the revision. Here is the summary of the update:
>   - We provides more details of each stage in data pipeline in sec.3.2, especially for the QA generation part.
>   - In each stage, we compare our approach with existing works. The main difference is that our pipeline includes new modules for data diversity (persona, domain-specific few-show examples) and aims to extract QA pairs grounded by the original doc instead of composing reasoning steps by a strong LLM.
>   - We further highlight the reference to appendix (e.g., prompt template, conversion examples) in main text to make it more clear to readers.
>
>   Feel free to let us know if you think some details should be further elaborated or some important literature is missing.

---

> > ### Author Response · Authors · 2025-11-26
> > **response to reviewer TSDn (2/2)**
> >
> > - **Q2. Ablation to show the diversity of the generated rl data and how it correlates with the final performance.**
> >
> >   As suggested by reviewer, we ran an ablation that only runs RL training by data from some domains to compare its performance with full domains. We  select a subset of domains and run RL on data only in those domains (with the same checkpoint) for 100 step (~50K data). We evaluate the model on MMLU-pro, BBH, GPQA.
> >
> >   |   | MMLU-pro | Big-Bench   | GPQA-D   |
> >   | --- | -------- | ---------   | -------- |
> >   | Qwen2.5-3B    | 37.8     | 41.2      | 20.8     |
> >   | Webscale-RL (all domains)     | **42.1** | **47.6**  | 22.8     |
> >   | remove 2 domains (coding, commerce)   | 41.8     | 47.5      | **23.2** |
> >   | remove 4 domains (..., healthcare, education)      | 41.9     | 45.9      | 22.7     |
> >   | remove 6 domains (..., natural science, lifestyle) | 40.5     | 44.8      | 21.2     |
> >
> >   The domain diversity can generally improve model performances on MMLU-pro and BBH which require multi-domain knowledge. Meanwhile, some domains (e.g., natural science) can be more important to specific benchmark (GPQA-diamond).
> >
> > We sincerely appreciate your effort again and we hope our responses can address your concerns.

---

### Meta-Review · Area_Chair_qs7d · 2026-01-03

**Summary:**

This paper proposes Webscale-RL, a scalable pipeline for converting large pretraining corpora into diverse, verifiable question-answer pairs for reinforcement learning. Using this pipeline, the authors construct a 1.2M example multi-domain dataset and show that RL training on this data substantially outperforms continual pretraining and strong data-refinement baselines, achieving comparable performance with dramatically fewer tokens.

**Reviewer Concerns:**

Reviewers raised concerns about novelty relative to prior work (e.g., NaturalReasoning), reliance on GPT-4.1 for data generation and verification, fairness of comparisons between RL and continual pretraining, and limited analysis of diversity and scalability. The rebuttal addressed most of these concerns with additional experiments and clarifications. It provided a controlled comparison against NaturalReasoning on matched domains, added domain ablation studies that show the impact of diversity to performance, and discussed cost and the use of alternative open-source models. Some limitations remain, such as lack of coding data, and lack of large model scaling, but these do not hurt the core contribution.

**Reviewer Scores:**

Reviewer cyfs: Unchanged (8), remained strongly positive after rebuttal.

Reviewer TSDn: Likely unchanged (6), with the concerns on diversity for generating verifiable answers addressed.

Reviewer giqw: Likely unchanged (6), with more generation details provided and evaluation fairness explanations.

Reviewer QNn9: Likely increased from 4 to 5--6, given the added comparison to NaturalReasoning and clarification on data generation and annotation pipeline relying on GPT-4.1.

---

### Decision · Program_Chairs · 2026-01-26

Accept (Poster)